# Identifying Predictors of Changes in Physical Activity Level in Adolescence: A Prospective Analysis in Bosnia and Herzegovina

**DOI:** 10.3390/ijerph16142573

**Published:** 2019-07-18

**Authors:** Vesna Miljanovic Damjanovic, Lejla Obradovic Salcin, Natasa Zenic, Nikola Foretic, Silvester Liposek

**Affiliations:** 1Clinic for Physical Medicine and Rehabilitation, University Hospital Mostar, 88000 Mostar, Bosnia and Herzegovina; 2Faculty of Health Sciences, University of Mostar, 88000 Mostar, Bosnia and Herzegovina; 3Faculty of Kinesiology, University of Split, 21000 Split, Croatia; 4University of Maribor, 2000 Maribor, Slovenia

**Keywords:** physical activity, substance misuse, puberty, sociodemographic variables

## Abstract

It is known that physical activity levels (PA levels) decline during adolescence, but there is a lack of knowledge on possible predictors of changes in PA levels in this period of life. This study aimed to prospectively investigate the relationship between sociodemographic and behavioral factors (predictors), PA levels and changes in PA levels in older adolescents from Bosnia and Herzegovina. The sample comprised 872 participants (404 females) tested at baseline (16 years of age) and at follow-up (18 years of age). Predictors were sociodemographic characteristics (age, gender, socioeconomic status, urban/rural residence, paternal and maternal education level) and variables of substance misuse (consumption of cigarettes, alcohol and illicit drugs). The PA level, as measured by the Physical Activity Questionnaire for Adolescents (PAQ-A), was observed as a criterion. Boys had higher PAQ-A scores than girls at baseline and follow-up. Paternal education levels were correlated with PAQ-A scores at baseline (Spearman’s R: 0.18, 0.15 and 0.14, *p* < 0.05, for the total sample, females and males, respectively) and at follow-up (Spearman’s R: 0.12, *p* < 0.01 for the total sample). Logistic regression, which was used to calculate changes in PA levels between baseline and follow-up as a binomial criterion (PA decline vs. PA incline), evidenced a higher likelihood of PA incline in adolescents whose mothers were more educated (OR: 1.29, 95% CI: 1.05–1.60) and who live in urban communities (OR: 1.56, 95% CI: 1.16–2.10). The consumption of illicit drugs at baseline was evidenced as a factor contributing to the lower likelihood of PA incline (OR: 0.36, 95% CI: 0.14–0.92). The negative relationship between illicit drug consumption and PA decline could be a result of a large number of children who quit competitive sports in this period of life. In achieving appropriate PA-levels, special attention should be placed on children whose mothers are not highly educated, who live in rural communities, and who report the consumption of illicit drugs. The results highlighted the importance of studying correlates of PA levels and changes in PA levels during adolescence.

## 1. Introduction

### 1.1. Physical Activity and Its Importance in Adolescence

Physical activity (PA) is an important public health concern [1,2]. Among the many health benefits, the most important are that PA helps prevent chronic conditions such as diabetes mellitus, cardiovascular diseases, and colon cancer [2,3,4]. Therefore, finding ways to reach the appropriate levels of PA is an important issue in modern society [5]. Although negative health-related consequences of an inadequate level of PA should be expected in late adulthood and older age, most health behavior patterns are established in childhood and adolescence, and therefore, PA promotion should begin at an early age [6]. In reaching appropriate levels of PA, it would be highly important to establish the factors associated with PA levels in different age groups. In brief, with the knowledge of specific sociodemographic and behavioral factors (predictors) and their influence on PA, public health authorities would be able to develop specific and targeted campaigns aimed at highlighting the factors of positive influence and/or controlling the factors of negative influence on PA [6,7,8,9].

In addition to the specific overall importance of reaching an appropriate level of PA in adolescence, the necessity of defining the factors that may influence PA in this period of life is also important because PA significantly declines during adolescence [10]. For example, studies confirmed the significant decline in various facets of PA in adolescent Portuguese girls [11] and in black and white girls in the United States (US) [12], while the most recent International Children’s Accelerometry Database (ICAD) project reported an average annual reduction in total physical activity of 3.5% to 4.7% relative to age five [13]. Not surprisingly, studies have already investigated the factors associated with PA levels worldwide. Among other factors that were targeted as potentially important determinants of PA levels, special attention was paid to various sociodemographic and behavioral variables [14,15].

### 1.2. Factors Associated with Physical Activity in Children and Youth

In one of the most comprehensive reviews of correlates of PA level in children and youth, Salis et al. reviewed the literature on a problem published before 1999 and identified several most important correlates of PA [14]. Apart from some logical and expected correlates of PA (i.e., participation in sport and physical educations classes), the authors highlighted other important factors associated with PA, including gender (with lower PA in girls), self-efficacy (with higher levels of PA in children with better self-efficacy), parental physical activity, parental education and parental support (all positively correlated with PA)) [14]. More recently, Van der Horst et al. reviewed the studies published between 1999 and 2005 that examined the correlates of children/adolescent PA and sedentary behavior [15]. In addition to the previously mentioned correlates described in the review of Sallis et al., authors highlighted the potential importance of certain behavioral variables, such as consumption of substances (substance use and misuse - SUM; cigarette smoking, illicit drug consumption), in explaining PA levels in adolescence [15,16,17].

Indeed, SUM in adolescence may be a factor of influence on PA levels, however, only few studies examined these issues with inconsistent findings. In short, cross-sectional study which comprised large sample of Greek youth (n = 5991) authors reported no significant correlation between smoking and “attitudes toward exercise” [18], and this is consistent with findings from study where authors observed US urban adolescent females aged 12–21 years [19]. Meanwhile, Nelson et al. in another US study evidenced lower likelihood of smoking for the adolescents clustered as being “physically active” (i.e., engaged in sports; “skaters/gamers”) when they were compared to their sedentary peers (“TV/video viewers”). Further, Peltzer reported a significant correlation between alcohol consumption and PA in 13–15 year old African children (e.g., more frequent alcohol consumption in those who had higher level of leisure time PA), but no association was evidenced between consumption of cigarettes and drugs, and PA [20]. In one of the rare prospective studies which examined the association between SUM and PA-changes in adolescence, cigarette smoking was identified as a predictor of decline in PA among adolescent girls in the US [12].

### 1.3. Study Problem and Aims

From the previous brief literature overview, it could be concluded that there is a clear necessity for further evaluation of the correlates of change in physical activity levels, and for such a purpose, prospective studies are needed [15]. Indeed, cross-sectional studies are important to determine the associations between the studied factors and PA, and in some cases, allow for the identification of the cause-effect relationship (i.e., gender is almost certainly a predictor of PA and not vice versa); but for more profound interpretations, prospective studies are needed. For example, the negative association between SUM and PA levels should be generated by at least two mechanisms: (i) the negative influence of SUM on physical capacities and therefore lower PA (where SUM is a predictor of lower PA) and (ii) specific social influence, such that children who are not included in sports and physical exercise are more likely to be in a specific sociocultural environment where SUM is often present (in this study, low PA is actually “a predictor” of SUM) [12,15,21,22].

Bosnia and Herzegovina is a country in southeastern Europe and is one of the former Yugoslav republics. After devastating wars in the early 1990s, the country is facing new challenges, including the necessity of developing effective public health campaigns aimed at reducing the health-related problems of modern societies, including issues related to PA levels in adolescence. To the best of our knowledge, only one study in a country territory has specifically investigated the problem of PA levels in adolescence, indicating inappropriate levels of PA in studied participants [23]. Furthermore, the problem of PA levels and the correlates of PA are generally understudied in the whole territory of the former Yugoslavia [24,25,26]. Also, to the best of our knowledge, there is no study that has prospectively investigated the predictors of changes in PA levels during adolescence in the southeastern European region. Therefore, to develop targeted and effective interventions that promote active lifestyles in these age groups, a better understanding of the relationships that may exist between different predictors and PA levels and PA changes is needed.

Collectively, previous studied evidenced regionally and culturally specific correlates of PA in different sample of adolescents [15,18,20]. Therefore, there is clear necessity for regionally and culturally specific investigation of correlates of PA-levels. Meanwhile, to the best of our knowledge, no study examined the changes in PA and correlates of PA in Bosnia and Herzegovina, while there is a clear lack of studies which have examined this problem in southeastern Europe. Globally, there is an evident lack of studies which prospectively investigated the factors potentially related to changes in PA-levels during the adolescence. Therefore, the aim of this study was to prospectively evaluate relationships that may exist among several sociodemographic variables (age, gender, socioeconomic status - SES, parental education, urban/rural environment), behavioral predictors (SUM variables), and PA levels in older adolescents from Bosnia and Herzegovina. Specifically, we examined predictors and PA levels when the studied adolescents were 16 years old on average and prospectively observed the influence of the studied predictors (i) on PA levels at the age of 18 and (ii) on the changes in PA levels over the observed period of life (between 16 and 18 years of age). Initially, we hypothesized significant correlations between predictors and changes in PA. Since previous studies indicated lower PA in females, adolescents of lower SES, and those who consume psychoactive substances [14,15], we hypothesized we would establish a negative influence of SUM and a positive influence of male gender, and higher SES on changes in PA levels.

## 2. Methods

### 2.1. Participants and Sampling

The participants in this prospective study were adolescents from three cantons in Bosnia and Herzegovina (B&H): The Tuzla County, the Herzegovina-Neretva County, and the Western Herzegovina County (Figure 1). At the study baseline (autumn 2016), participants were attending their 3rd year of high school and were 16 years old on average (1st wave). The follow-up testing commenced at the end of the 4th year of high school (late spring 2018), approximately 20 months after baseline testing (2nd wave). At baseline, we tested 934 participants, but in this study, we included 872 participants (404 females) who were tested at both baseline and follow-up. The drop-out rate was therefore 7%. With a theoretical sample of approximately 11,300 adolescents aged 16 years and a level of significance of *p* < 0.05, the required sample size was 369 participants.

The sampling was performed over several phases. First, all the high schools in the territories of two selected counties were stratified into two groups according to their size. Second, we randomly selected one-third of the 3rd–year classes from two defined groups. Since some high school programs end in the 3rd year, we selected only classes included in 4-year high school programs (altogether 36 classes). The investigators then visited the schools during the first week of the academic year and distributed the consent forms to all children in selected classes. The following week, the baseline testing was performed, involving children who personally agreed to participate and whose parent(s) signed the consent form. Although the testing was anonymous, to pair the responses at two testing waves, participants were instructed to self-select anonymous codes for identification purposes. Specifically, they were instructed to use the last three digits of their e-mail password to remember the code more easily. Testing was performed in groups of at least 10 participants, and at the beginning, the examiners informed the participants of the study purpose and aims, that testing was absolutely voluntary and anonymous, that there was no intention to connect responses to a single person and that they could leave the questionnaire form and/or some questions unanswered. Testing lasted approximately 15 minutes, and participants were then asked to place the form in the provided envelope, seal it and put the envelope in the closed box, which was opened the day after the testing. The procedure and study fulfilled ethical guidelines and was approved by University of Mostar, Faculty of Medicine (Mostar, Bosnia and Herzegovina, and University of Split, Faculty of Kinesiology (Split, Croatia).

After follow–up testing, we calculated an analysis of attrition bias, which evidenced no significant difference in baseline PA levels between groups of participants (e.g., those tested over both testing waves and those who dropped out). However, significantly more females than males remained in the study (chi-square = 12.11, *p* < 0.01), which may be attributable to the fact that previous studies have regularly evidenced more school absences among adolescent boys than among girls [27]. Additionally, the intracluster correlation (IC) for the baseline PA level (with the schools as clusters) showed appropriate within-school variance (IC = 0.04 and 0.05 for baseline and follow-up) [28,29].

### 2.2. Variables

The variables in this study included participants’ sociodemographic characteristics, SUM, and PA levels. The sociodemographic characteristics included gender (male-female-prefer not to answer), age (in years), paternal and maternal education level (elementary school-high school-college degree-university degree), place of residence (which was later grouped and observed as “urban” and “rural”), and the socioeconomic status of the family (under average-above average).

SUM variables included questions on cigarette smoking, alcohol drinking and illicit drug consumption. Cigarette smoking was addressed with one question with four possible answers (never smoked-quit-from time to time but not daily-daily smoking), but participants were later grouped into “nonsmokers” (first two responses) and “smokers” (last two responses). Alcohol drinking was questioned using the Alcohol Use Disorders Identification Test (AUDIT), with theoretical scores ranging from 0 to 40 [30]. For the purposes of this study and statistical analyses (more detailed information later in the paper), we observed raw scores, but participants were also divided into two groups with 11 as a cut-off score (e.g., “harmful drinkers” vs. “nonharmful drinkers”). The scale for illicit drug consumption included questions about the consumption of 11 different illicit drugs (i.e., marijuana, hashish, heroin, cocaine, sedatives, and most party drugs [e.g., ecstasy, amphetamines]), with a seven-point range of consumption for each drug (ranging from “never”, “ever tried”, and “more than once” to “40 times and more”). Participants were later categorized as “drug users” and “nonusers”, as previously suggested [27].

PA levels were evidenced by the Physical Activity Questionnaire for Adolescents (PAQ-A), a reliable and valid questionnaire form that was previously translated and studied for reliability and validity in samples of adolescents, including those from former Yugoslav countries [23,31,32,33]. The questionnaire consists of nine items asking participants to provide seven-day self-report recall, with a final theoretical score ranging from 0 (minimum) to 5 (maximum PA level). The first 8 items are scored on a 5-point scale and include questions on various types of PA (i.e., PE activity during physical education classes, sports, free play, active transportation), and therefore the PAQ-A provides a summary PA score derived from eight items (the 9th item does not contribute to the overall score but rather is used only for selection [i.e., illness, injury]). To identify changes in PA during the observed period of time, the difference between the PAQ-A results obtained at the study baseline and those obtained at follow-up was calculated for each participant (e.g., PAQ-A at follow-up “minus” PAQ-A at baseline). Consequently, we were able to identify two categories of adolescents: (i) those whose PA levels declined and (ii) those whose PA levels inclined during the study period. Such categorization was later used as the criterion for logistic regression (e.g., PA incline vs. PA decline; please see statistical analyses for more details).

### 2.3. Statistics

The Kolmogorov-Smirnov test was used to check the normality of the distribution. As a result, means and standard deviations were calculated for numerical variables (i.e., PAQ-A scores, AUDIT), while percentages and frequencies were reported for the remaining variables (Appendix A).

To demonstrate the differences in PAQ-A scores between boys and girls, and between children residentially located in urban and rural environments, the independent samples t-test was performed. Additionally, differences between groups were identified by calculation of Effect Size differences (ES) with 95% Confidence Intervals (95% CI), with modified qualitative descriptors (trivial ES: < 0.2; small ES: 0.21–0.60; moderate ES: 0.61–1.20; large ES: 1.21–1.99; and very large ES: > 2.0 [34]).

Furthermore, the dependent samples t–test and ES were calculated to identify differences in PAQ-A scores with regard to changes that occurred from baseline to follow–up.

Because of the differences in parametric/nonparametric nature of the variables the associations between ordinal variables (e.g., parental education level, SES) and raw scores of PAQ-A were calculated by Spearman’s rank-order correlation coefficients. To identify the associations between predictors and binomial criterion (PA-incline vs. PA-decline) logistic regressions was calculated, with Odds Ratios (ORs) and corresponding 95% CI reported. The model fit was checked using the Hosmer Lemeshow test (statistically significant test indicates that the model does not adequately fit the data). A *p*-value of 0.05 was applied, and the statistical package Statistica ver. 13.0 (Statsoft, Tulsa, OK) was used for all calculations.

## 3. Results

Boys had higher PAQ-A scores than girls at study baseline (2.75 ± 0.71 and 2.14 ± 0.56, t-test: 13.95, *p* < 0.01, respectively; moderate ES [ES: 0.95, 95% CI: 0.81–1.1]) and follow-up (2.52 ± 0.77 and 1.94 ± 0.95, t-test: 12.42, *p* < 0.01, respectively; moderate ES [ES: 0.67, 95% CI: 0.53–0.81]). The PA, as evidenced by PAQ-A scores, significantly declined over the study period in the total sample (from 2.46 ± 0.71 to 2.25 ± 0.73, t-test: 9.34, *p* < 0.01; small ES [(ES: 0.33]), and when observed separately for boys (from 2.75 ± 0.71 to 2.52 ± 0.77, t-test: 6.71, *p* < 0.01; small ES [(ES: 0.24]), and for girls (from 2.14 ± 0.56 to 1.94 ± 0.55, t-test: 7.16, *p* < 0.01; small ES [ES: 0.25]) (Figure 2A).

There was no significant difference in PAQ-A scores between children living in urban and rural environments (t-test: 0.83 and 0.48, *p* > 0.05, for baseline and follow–up, respectively) (Figure 2B).

Paternal education levels were significantly correlated with PAQ-A at baseline (Spearman’s R: 0.18, 0.14 and 0.14, *p* < 0.05, for the total sample, females and males, respectively) and follow-up but only for the total sample of participants (Spearman’s R: 0.12, *p* < 0.01). For total sample and girls, maternal education levels were correlated with changes in PA levels, with higher maternal education in children whose PA increased during the course of the study (Spearman’s R: 0.07 and 0.10, *p* < 0.05, for total sample and girls, respectively) (Table 1).

When PA changes were observed as binomial criterion in logistic regression (PA decline [coded as “1”] vs. PA incline [coded as “2”]), significant relationships between predictors and criterion were evidenced for rural/urban environment, maternal education, and the consumption of illicit drugs. Specifically, a higher likelihood of PA incline during the study period was found for adolescents who lived in urban environment (OR: 1.56, 95% CI: 1.16–2.10, *p* = 0.01), and those whose mothers were more educated (OR: 1.29, 95% CI: 1.05–1.51, *p* = 0.02). Oppositely, the consumption of illicit drugs at baseline was evidenced as a factor of a lower likelihood of PA incline during the study period (OR: 0.37, 95% CI: 0.14–0.92, *p* = 0.03). The remaining variables were not significantly correlated with PA incline/decline (OR [95% CI]: Age: 0.99 [0.55–1.04], Gender: 0.86 [0.66–1.13], Socioeconomic status: 1.18 [0.69–2.05], Paternal education: 1.13 [0.93–1.39], Smoking: 0.96 [0.86–1.07], AUDIT: 0.99 [0.97–1.02]; all *p* > 0.05) (Figure 3). The Hosmer-Lemeshow test indicated that logistic regression model fit the data appropriately (Chi square: 12.81, *p* = 0.12) 

## 4. Discussion

There are several important findings of this study. First, the PA levels of the adolescents studied significantly declined during the observed period, irrespective of gender. Next, the PA levels at baseline were positively correlated with fathers’ educational levels. The higher maternal educational level and living in urban environment predicted the PA increase over the observed period of time. Finally, the consumption of illicit drugs at baseline was negatively correlated with changes in PA, with a lower likelihood of PA incline in adolescents who used illicit drugs at baseline. Therefore, our initial study hypothesis on significant associations between sociodemographic status and SUM with PA was generally confirmed.

### 4.1. Physical Activity Level and Changes in Studied Adolescents

The PA levels of adolescents of B&H are rarely studied, and to the best of our knowledge, only one study has examined PA levels of adolescents using the same scale as we used in this study (the PAQ-A). Specifically, Pojskic and Eslami recently noted somewhat higher levels of PA in 10- to 14-year-old children from B&H (e.g., 2.91 ± 0.63 and 3.16 ± 0.65 in girls and boys, respectively) [23], which corresponds with findings from a report on same-aged Croatian children (2.90 ± 0.61 for total sample) [26]. Taking into account that (i) we studied older children than our colleagues who noted minimally higher levels of PA, (ii) there was an evident decrease in PA during adolescence, and (iii) there were consistently higher levels of PA in boys than in girls, our PA levels are actually consistent with findings from previous reports [23,26]. Also, a study on 15-year-old Greek adolescents demonstrated almost identical PAQ-A values, as we observed in this study (2.97 ± 0.35 and 2.58 ± 0.31 for boys and girls, respectively) [35].

The decrease in adolescents’ PA levels is a global problem, and it is recognized as an important health-compromising behavior in modern society [36]. More precisely, while PA levels are generally observed as adequate during childhood, they dramatically decline during adolescence for numerous reasons (i.e., an increase in scholastic duties, “screen time”, a decline in active transport, and an increase in overweight, which consequently reduces PA). Consequently, studies around the globe have confirmed inadequate PA levels in late adolescence [37]. On the other hand, our results suggesting a similar decline in PA in boys and girls are in certain disagreement with studies performed thus far, in which authors have mostly indicated girls to be a high-risk group for “low physical activity”, with an either implicit or explicit assumption that boys are at least partially protected against such negative trends [36,37]. More specifically, while our results support the previous findings of higher PA levels in boys than in girls, according to the trends observed in this study (e.g., similar decline in both boys and girls between 16 and 18 years of age), it seems that the background of higher PA in adolescent boys should be found in younger ages.

### 4.2. Sociodemographic Factors and Physical Activity in Adolescence

We found a generally positive influence of parental education on children’s PA, but we also found a specific influence of paternal and maternal educational levels on the PA status of their children. Before discussing these specifics, we will first broadly discuss the general mechanisms of established relationships. The possible associations between parental education and child health have been frequently investigated [38,39,40]. In early studies, the associations between these factors were mostly observed from the perspective of overall hygiene, quality of nutrition and child mortality, but with the growing emergence of other public health issues, the focus has moved toward SUM and PA [40,41,42]

Interestingly, more than 20 years ago, authors from Finland confirmed the positive influence of both maternal and paternal educational levels on PA levels in 9-year-olds [43], but their results were later not consistently confirmed. For example, in a more recent study, the international group of authors examined the associations between parental education and PA in children from seven European countries and found generally positive but, in most cases, gender-specific influence of parental education on the PA levels of their children [41]. In a more recent study performed on Ghanian adolescents, the authors did not confirm a significant effect of parental education on the PA levels of their children [44], while other international investigations highlighted even negative associations between parental education and child PA for lower economic status countries [45].

Our results confirmed a positive association between paternal educational levels and PA at study baseline in both boys and girls. Although numerical values of the correlation coefficients were not high, this specific association is interesting because identical “parentally specific influence” is evidenced in one nearby country: Hungary [41]. Although Hungary does not share a border with B&H, these two countries share similar histories, traditions, and even scholastic systems, since a significant part of B&H was part of the Austro-Hungarian Empire, including two of three counties studied herein [46]. This finding altogether confirms the conclusions found in previous studies, in which authors highlighted the necessity of the specific evaluation of relationships between parental education and PA with regard to the culture, traditional background and development of the observed country/region [45].

We may say that in B&H, both parents share similar responsibilities, and therefore, both parents similarly influence the educational and behavioral development of their children. This is clear even from our data on the educational status of both parents (i.e., with 8.7% of fathers and 7.0% of mothers with university degrees (Appendix A), there was no evident difference in the education levels of parents). However, we cannot ignore the specific parental influence on different facets of a child’s life. In brief, while mothers are culturally “more responsible” for children’s educational achievement, fathers are more influential in regard to sports participation, and these “educational specifics” are not characteristic only for B&H or the studied region but rather are global paradigms [47,48]. While most of the PA levels at baseline in B&H adolescents are indicative of active participation in competitive sports, the positive association between paternal education and PA at baseline is understandable. Meanwhile, previous studies have evidenced a significant decrease in participation in competitive sports in this period of life, which simply statistically reduced paternal influence on children’s PA levels [49,50].

While fathers and their educational levels seemed to be more influential in children’s PA in earlier adolescence, maternal education significantly predicted the changes in PA that occur between 16 and 18 years of age. Specifically, a higher likelihood of increased PA is evidenced among children whose mothers were more educated. Since this is one of the rare studies that examined this issue specifically while observing (i) paternal and maternal educational levels separately and (ii) changes in PA between observed testing waves, the following discussion on a problem cannot be supported by previous findings on similar associations but rather is based on information from other fields. In short, the authors share the opinion that the background of the influence of maternal education on PA changes over the observed period is actually the result of the strong influence of mothers on children’s educational achievements [51]. With the decrease in participation in competitive sports in late adolescence, PA levels logically become more of a result of personal awareness of the health-related benefits of PA and the overall importance of PA. Therefore, the PA itself, and particularly the incline in the PA at this age, should be observed as a result of proper health education. The positive influence of maternal education on PA increases in this period of life should therefore be observed as a natural consequence of the known positive influence of maternal education on general educational achievements of children [51,52,53].

The adolescents from urban communities were more likely to increase the PA during the course of the study. This finding can be logically explained from the perspective of higher availability of recreational and fitness centers in urban, than in rural communities [54,55]. Therefore, adolescents who live in urban centers simply have better possibilities to be physically active than their peers who live in rural communities. If we take into consideration that everyday PA decreases globally, it is not surprising that studies confirm increased prevalence of overweight, particularly in rural children [56].

### 4.3. Substance Use and Misuse and Physical Activity in Adolescence

Only a few studies have examined the problem of association between SUM and PA in adolescence with inconsistent findings [12,15,21,22]. In our study, cigarette smoking was not identified as a predictor of changes in PA for the studied adolescents. Because of the evident lack of consensus with regard to the association between smoking and PA, we may say that our results are not surprising [18,19]. However, since one of the rare prospective studies performed on a problem-confirmed decline in PA in adolescent smokers in the US, it would be important to provide a possible interpretation of the difference between our findings and those of our respected colleagues [12].

Most likely, the high prevalence of smoking in B&H adolescents should be considered one of the most likely explanations for the lack of association between smoking and PA. In short, previous studies conducted in this territory confirmed alarming figures on smoking prevalence among adolescents, and with approximately 30% of smokers, these alarming figures are confirmed in this study [27,57]. Moreover, contrary to most of the worldwide studies in which authors have frequently confirmed a negative association between smoking and sports participation, the studies conducted in the territory of the former Yugoslavia (including B&H) did not confirm such findings, which has been frequently explained by the social acceptance of smoking in the territory [57,58]. As a result, it is reasonable to expect that such figures are actually translated to a lack of association between cigarette smoking and PA changes in late adolescence.

Previous investigations that specifically observed the consumption of different illicit drugs (i.e., marijuana, hashish) did not confirm the correlations between this type of SUM and PA in adolescence [18,19]. Such results are in certain disagreement with our findings, in which we evidenced a lower likelihood of positive changes in PA levels among adolescents who reported the consumption of illicit drugs at baseline. However, even in our study, illicit drug consumption was not correlated with illicit drug consumption in the cross-sectional analyses (at baseline and follow-up). Because of the evident lack of studies which has prospectively examined the influence of illicit drug consumption on changes in PA in adolescence, we may say that our results are pioneering to some extent. Therefore, some specific results that were obtained in the past are helpful in explaining the findings.

The illicit drug consumption is found to be specifically associated with sports participation in adolescents from the territory, and some studies have even identified an increased risk of the consumption of illicit drugs among athletic adolescents [59]. The age of 16–17 years (the baseline wave in this investigation) is known to be “critical” with regard to participation in competitive sports, and studies have regularly confirmed a significant rate of drop-out from a sport during this period of life [49,59]. Therefore, it is logical that PA levels at this age decline as well. Collectively, it results even in association between illicit drugs consumption at baseline, and PA-decline during the course of the study.

### 4.4. Limitations and Strengths

The first limitation of this study comes from the fact that the PA levels were not objectively measured but rather were evidenced throughout the questionnaire. However, the questionnaire used was reported to be a reliable and valid measuring tool. Additionally, because of the (i) relatively large sample of participants, (ii) testing of the SUM data, and (iii) study design (two-year follow up), the usage of more accurate measurement tools (i.e., pedometers, accelerometers) was limited. Furthermore, there is a certain possibility that participants may lean toward socially desirable answers, especially regarding the questioning of SUM. However, we believe that strict anonymity and our experiences from previous studies reduced this possibility. Additionally, this study did not involve adolescents from the territory of the whole country, which clearly limits the generalizability of the results, but we believe that the fact that we analyzed two specific regions (i.e., “continental” and “Mediterranean” Counties) at least partially reduces the impact of this limitation.

This is one of the rare studies that investigated correlates of PA in adolescence and is probably the first prospective investigation on a problem in southeastern Europe. Furthermore, a relatively large sample of participants and a high retention rate (e.g., small drop-out number) are important strengths of the investigation. Finally, this is probably the first study that which prospectively investigated the predictors of PA changes in late adolescence in southeastern Europe. Given the global consensus on the importance of the problems studied, we hope that our results will contribute to knowledge in this field and inspire further research.

## 5. Conclusions

Our results confirmed that PA levels are lower in girls than in boys. However, a prospective study design allowed us to identify that girls are not at higher risk of PA decline between 16 and 18 years of age. Collectively, these results highlight the necessity of investigating the trends and changes in PA levels in earlier adolescence because differences between boys and girls evidently occur in an earlier period of life. At the same time, the focus should be moved toward different types of PA in which adolescents are involved before the age of 16 years.

The study revealed specific associations between parental education and PA levels in adolescents from B&H. While adolescents whose fathers were more educated had higher PA levels when they were 16 years of age, maternal education was positively correlated with PA incline over the next two years. Therefore, we may conclude that fathers have a stronger influence on PA at an earlier age, most likely because of their stronger influence on children’s active sports participation. On the other hand, mothers are probably more influential on PA in later adolescence, mostly because of the importance they place on children’s health-related behaviors. However, these issues should be more precisely investigated in future (preferably qualitative) studies, in which participants would be specifically asked about the possible background of the relationships established in this study. Since our results confirmed a higher likelihood for PA incline in urban adolescents, future studies should be specifically focused on children who live in rural communities.

The consumption of illicit drugs at baseline, when adolescents were 16 years old, was found to be negatively related to changes in PA levels during the next two years. Therefore, it seems that adolescents who initiate drug consumption at an earlier age are in particular danger of developing other negative habits that can directly and/or indirectly harm their overall health status. There is evidence that such problems are specific to adolescents who were involved in competitive sports at an earlier age and who quit sports at ages 16–17 years. In future studies, this issue should be specifically investigated, and if the previously described conclusions are confirmed, then specific and targeted campaigns that focus on former athletes should be urgently developed.

## Figures and Tables

**Figure 1 ijerph-16-02573-f001:**
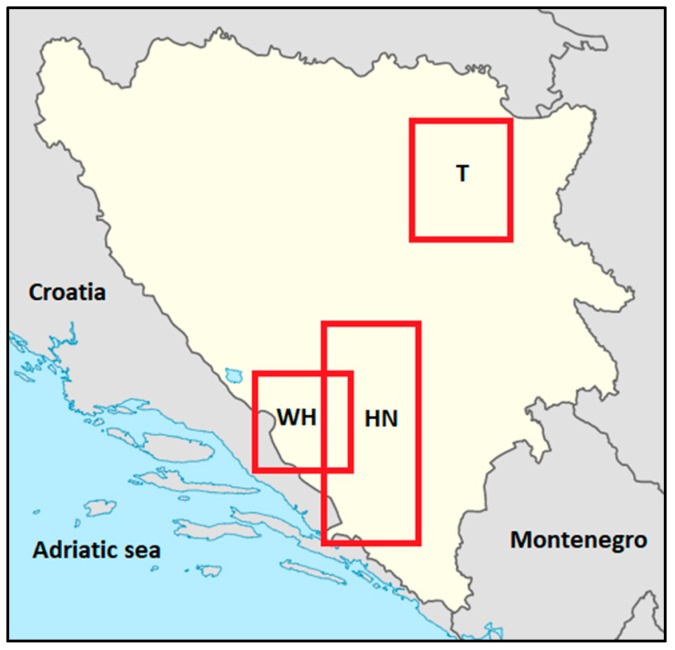
Bosnia and Herzegovina with the approximate location of the Counties where the sample was drawn from (HN-Herzegovina Neretva County, WH-Western Herzegovina County, T-Tuzla County).

**Figure 2 ijerph-16-02573-f002:**
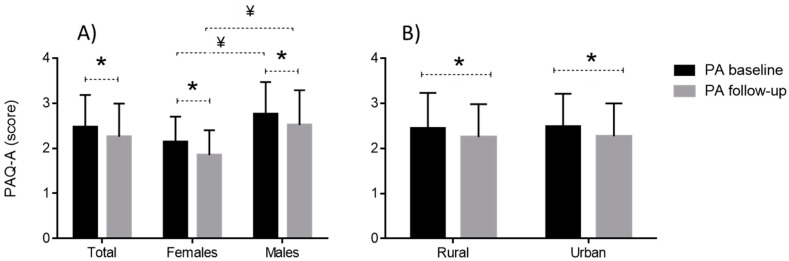
Descriptive statistics (presented as Means+Standard Deviations) for the physical activity level as measured by Physical Activity Questionnaire for Adolescents (PAQ-A) at baseline and follow–up, with differences between groups and within groups (* indicates significant [*p* < 0.01] within–group differences obtained by t-test for dependent sample; ^¥^ indicates significant [*p* < 0.01] between group differences obtained by t-test for independent samples).

**Figure 3 ijerph-16-02573-f003:**
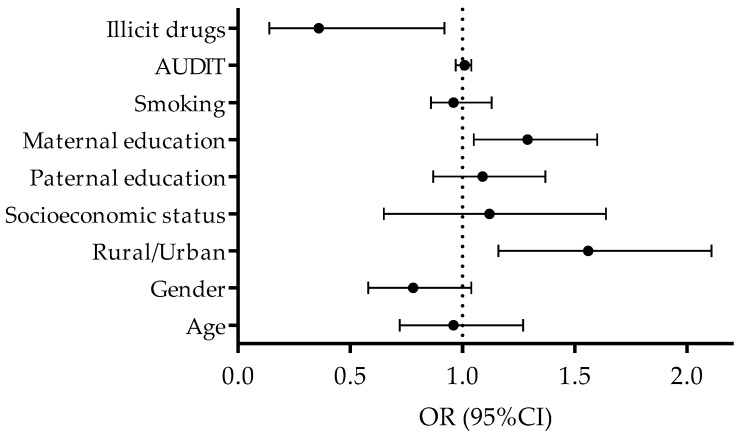
Correlates of decline/incline in PA level over the course of the study (OR-Odds Ratio, 95% CI-95% Confidence Interval).

**Table 1 ijerph-16-02573-t001:** Spearman’s rank order correlations between sociodemographic variables and variables of substance use and misuse with physical activity as measured by Physical Activity Questionnaire for Adolescents (PAQ-A) in the studied adolescents.

	PAQ-A Baseline			PAQ-A Follow-Up			PAQ-A Changes		
	Total	Girls	Boys	Total	Girls	Boys	Total	Girls	Boys
**Age**	−0.13 *	−0.06	−0.14 *	−0.06	−0.02	−0.08	−0.07 *	−0.05	−0.08
**Socioeconomic Status**	0.01	0.01	−0.04	−0.03	−0.04	−0.05	0.03	0.05	0.03
**Paternal Education**	0.18 *	0.14 *	0.14 *	0.12 *	0.09	0.07	0.06	0.06	0.05
**Maternal Education**	0.07	0.04	0.01	0.03	−0.04	−0.01	0.07 *	0.10 *	0.05
**Smoking**	−0.02	−0.08	−0.05	−0.01	−0.07	−0.02	−0.01	0.01	−0.02
**AUDIT**	0.13 *	0.02	0.06	0.13 *	0.03	0.06	0.02	0.03	0.02
**Illicit Drugs**	−0.01	0.00	0.04	−0.01	0.08	−0.03	−0.02	−0.12 *	0.08

Legend: PAQ-A baseline-physical activity at baseline, PAQ-A follow-up-physical activity at follow up, PAQ-A changes-changes in physical activity over the curse of the study, AUDIT-alcohol drinking as measured by Alcohol Use Disorders Identification Test (* indicates statistical significance of *p* < 0.05).

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
