# Peer review of "Identifying Predictors of Changes in Physical Activity Level in Adolescence: A Prospective Analysis in Bosnia and Herzegovina"

_ijerph, 2019, doi:10.3390/ijerph16142573_

Round 1

Reviewer 1 Report

This study examines the relationships between SUM variables and PA levels in older adolescents from Bosnia and Herzegovina. The topic of the study is interesting. It is well written. The results contribute to knowledge in the field. I only have a suggestion. According to the Physical Activity Questionnaire, PA levels are a continuous variable; however, PA levels were recoded into a dichotomous variable (a logistic regression analysis was performed). Forcing a continuous variable into a categorical framework leads to a considerable loss of information. The authors may need to use a multiple regression analysis rather than a logistic regression analysis.

Author Response

Reviewer 1

This study examines the relationships between SUM variables and PA levels in older adolescents from Bosnia and Herzegovina. The topic of the study is interesting. It is well written. The results contribute to knowledge in the field. I only have a suggestion. According to the Physical Activity Questionnaire, PA levels are a continuous variable; however, PA levels were recoded into a dichotomous variable (a logistic regression analysis was performed). Forcing a continuous variable into a categorical framework leads to a considerable loss of information. The authors may need to use a multiple regression analysis rather than a logistic regression analysis.

RESPONSE: Thank you for recognizing the quality in our paper. With regard to your suggestion (usage of the multiple regression instead of logistic regression) we may say that we had the same idea originally. However, the problem was with “predictors”, which were mostly ordinal (parental education, socioeconomic status, smoking, illicit drugs consumption), or even nominal (gender, rural/urban community). Actually, exclusively “age” and raw AUDIT score were parametric in its nature, while “age” was not normally distributed. For that reason we originally decided not to use “classic” regression analysis, but calculated difference by t-test (for nominal variables; please see Figure 2), and Spearman’s rank order correlation (for ordinal variables and PA criteria). Also, we must mention that logistic regression was actually not calculated on “binomialized” PA scores, but exclusively on data about “PA-incline”, and “PA-decline”. However, according to your suggestion we have learned that this was not sufficiently explained in the manuscript, and therefore these issues are explained in more details in the revised version of the manuscript. Text reads: “The Kolmogorov-Smirnov test was used to check the normality of the distribution. As a result, means and standard deviations were calculated for numerical variables (i.e. PAQ-A scores), while percentages and frequencies were reported for the remaining variables (Supplementary table 1). To demonstrate the differences in PAQ-A scores between (i) boys and girls and between (ii) children residentially located in urban and rural environments, the independent samples t-test was performed. Additionally, differences between groups were identified by calculation o Effect Size differences (ES) with 95% Confidence Intervals (95%CI), with modified qualitative descriptors (trivial ES: < 0.2; small ES: 0.21-0.60; moderate ES: 0.61-1.20; large ES: 1.21-1.99; and very large ES: > 2.0 [34]. Furthermore, the dependent samples t-test was used to identify significant differences in PAQ-A scores with regard to changes that occurred from baseline to follow-up. Because of the differences in parametric/nonparametric nature of the variables the associations between ordinal variables (e.g., parental education level, SES) and raw scores of PAQ-A for baseline and follow-up were calculated by Spearman’s rank-order correlation coefficients. To identify the associations between predictors and binomial criterion (PA-incline vs. PA-decline) logistic regressions was calculated, with Odds Ratios (ORs) and corresponding95%CI reported. The model fit was checked by Hosmer Lemeshow test (statistically significant test indicates that the model does not adequately fit the data). A p-value of 0.05 was applied, and the statistical package Statistica ver. 13.0 (Statsoft, Tulsa, OK) was used for all calculations.” (please see Statistics subsection).

Staying at your disposal.

Authors

Reviewer 2 Report

This study aimed to prospectively investigate the relationship between sociodemographic and behavioural factors (predictors), PA levels and changes in PA levels in older adolescents from Bosnia and Herzegovina. The article has an impressive sample and explores a potentially interesting topic; I do however have a number of suggestions for this paper that I believe should be addressed before it can be considered for publication.

General comments:

There are frequent grammatical and typographical errors present throughout the paper, which mar the reader’s understanding of the paper and also cast doubts on the authors’ own understanding of the main purpose of this study. The paper should, therefore, be thoroughly revised to remove all grammatical and typographical errors, preferably by a native English-speaking person.

Introduction:

1-      The introduction of the paper was very descriptive, it did not situate the current study in literature or highlight what the gap in the literature is that this study is trying to address. In fact, physical activity is like a “pill”, especially in adolescent. As authors know both scientific papers and several European and World reports.  My first concern here is that authors reported the percentage’s from these reports; 

2-      What are the theoretical and empirical rationales that support the present study? What are gaps in the literature, regarding this topic?

3-      Why you believe in these hypotheses? The hypotheses of the present study must be better described, in terms of why you believe?

4-      My last concern is related to introduction structure. In other words, if authors split the discussion in different sub-headings, could adopt the same strategy for the introduction. I believe that this way could be easier for readers to follows the paper.

Methodology:

In general, methodology is well defined and described. However, the statistical analyses do not convince me, since the following topics:

1.      The authors used independent sample t-test in order to compare groups. Please report the partial eta-square and defined the cut-off values form the effect size based on Hopkins (2009) or Cohen (1988; 1992);

2.      For my surprise, the authors used logistic regression. However, logistic regression does not allow identifying either the global adjustment of performed models.  In line with that, I suggest SEM analysis (for manifest variables) or Path analysis whether models are saturated (Kline, 2016). These kinds of analysis are more robustness, especially, because authors recruited a large sample.

3.      It is unclear if authors based their statistics differences on p-value or confidence interval. If authors based their interpretation on CI-95% please reported the reference, which justifies it.

4.      In line 196 (page 5), authors said the following: “A p-value of 95% was applied”. Please note that p-value is not a 95%. Otherwise, p-value represents the level of probability of rejecting the null hypothesis. In this case must be 0.05, when the confidence interval is 95%.

Results

Results heading is well reported and described. However, based on my previous comments, they can be modifying.

Discussion

In general the discussion is well done conducted.

Author Response

REVIEWER 2

This study aimed to prospectively investigate the relationship between sociodemographic and behavioural factors (predictors), PA levels and changes in PA levels in older adolescents from Bosnia and Herzegovina. The article has an impressive sample and explores a potentially interesting topic; I do however have a number of suggestions for this paper that I believe should be addressed before it can be considered for publication.

RESPONSE: Thank you for recognizing the potential of our manuscript. Please find bellow the responses to each of your comments.

General comments:

There are frequent grammatical and typographical errors present throughout the paper, which mar the reader’s understanding of the paper and also cast doubts on the authors’ own understanding of the main purpose of this study. The paper should, therefore, be thoroughly revised to remove all grammatical and typographical errors, preferably by a native English-speaking person.

RESPONSE: We have tried to follow all your comments and amended the manuscript accordingly. For details on each of your comments and how we responded please see responses.

Introduction:

1-           The introduction of the paper was very descriptive, it did not situate the current study in literature or highlight what the gap in the literature is that this study is trying to address. In fact, physical activity is like a “pill”, especially in adolescent. As authors know both scientific papers and several European and World reports.  My first concern here is that authors reported the percentage’s from these reports;

RESPONSE: Thank you for your suggestion. The Introduction now includes several new reports on a problem of PA in adolescence and issues related to lack of PA in this age group. Specifically, in this version of the manuscript we added a complete new paragraph about the main problem of the research (e.g. association which may exist between SUM and PA in adolescence). Text reads: “Indeed, SUM in adolescence may be a factor of influence on PA levels, however, only few studies examined these issues and findings were not consistent. In short, cross-sectional study which comprised large sample of Greek youth (n = 5991) authors reported no significant correlation between smoking and “attitudes toward exercise” in Greek youth [18], and this is consistent with findings from study where authors observed US urban adolescent females aged 12-21 ears [19]. Meanwhile, Nelson et al. in another US study evidenced lower likelihood of smoking for the adolescents clustered as being “physically active” (i.e. engaged in sports; “skaters/gamers”) when they were compared to their sedentary peers less (“TV/video viewers”). Further, Peltzer reported significant correlation between alcohol consumption and PA in 13-15 year old African children (e.g. more frequent alcohol consumption in those who had higher level of leisure time PA), but no association was evidenced between consumption of cigarettes and drugs, and PA [20]. In one of the rare prospective studies which examined the association between SUM and PA-changes in adolescence, cigarette smoking was identified as a predictor of decline in PA among adolescent girls in the US [12]”.

2-           What are the theoretical and empirical rationales that support the present study? What are gaps in the literature, regarding this topic?

RESPONSE: We must agree that “the study problems” was not sufficiently explained. Therefore, in this version of the manuscript we tried to explain it in more details. Text reads: “From the previous brief literature overview, it could be concluded that there is a clear necessity for further evaluation of the correlates of change in physical activity levels, and for such a purpose, prospective studies are needed [15]. Indeed, cross-sectional studies are important to determine the associations between the studied factors and PA, and in some cases, allow for the identification of the cause-effect relationship (i.e., gender is almost certainly a predictor of PA and not vice versa); for more profound interpretations, prospective studies are needed. For example, the negative association between SUM and PA levels should be generated by at least two mechanisms: (i) the negative influence of SUM on physical capacities and therefore lower PA (where SUM is a predictor of lower PA) and (ii) specific social influence, such that children who are not included in sports and physical exercise are more likely to be in a specific sociocultural environment where SUM is often present (in this study, low PA is actually “a predictor” of SUM) [12,15,21,22].” (please see lines 90-100)

3-           Why you believe in these hypotheses? The hypotheses of the present study must be better described, in terms of why you believe?

RESPONSE: In this version of the manuscript the part of the text where we specified hypotheses of the study is systematically rewritten and we believe that the background for specified hypotheses is now more evident. Text reads: Initially, we hypothesized significant correlations between predictors and changes in PA. Since previous studies indicated lower PA in females, adolescents of lower SES, and those adolescents who consume psychoactive substances [14,15], we hypothesized we would establish a negative influence of SUM and a positive influence of male gender, and higher SES on changes in PA levels.” (please see lines 125-129)

4-           My last concern is related to introduction structure. In other words, if authors split the discussion in different sub-headings, could adopt the same strategy for the introduction. I believe that this way could be easier for readers to follows the paper.

RESPONSE: We are particularly grateful for this suggestion. Accordingly, the Introduction is now “divided” into subheading: 1.1. Physical activity and health in adolescence; 1.2. Correlates of physical activity in children and youth; 1.3. Study problem and aims (please see subheadings of the Introduction). Thank you.

Methodology:

In general, methodology is well defined and described. However, the statistical analyses do not convince me, since the following topics:

1.           The authors used independent sample t-test in order to compare groups. Please report the partial eta-square and defined the cut-off values form the effect size based on Hopkins (2009) or Cohen (1988; 1992);

RESPONSE: As you suggested, the Effect Size differences with corresponding 95%CI intervals is calculated (specified in the Statistics subsection; text reads: “To demonstrate the differences in PAQ-A scores between (i) boys and girls and between (ii) children residentially located in urban and rural environments, the independent samples t-test was performed. Additionally, differences between groups were identified by calculation o Effect Size differences (ES) with 95% Confidence Intervals (95%CI), with modified qualitative descriptors (trivial ES: < 0.2; small ES: 0.21-0.60; moderate ES: 0.61-1.20; large ES: 1.21-1.99; and very large ES: > 2.0 [34]”) and results are briefly presented in the Results section (text reads: Boys had higher PAQ-A scores than girls at study baseline (2.75 ± 0.71 and 2.14 ± 0.56, t-test: 13.95, p < 0.01, respectively; moderate ES [ES: 0.95, 95%CI: 0.81-1.1]) and follow-up (2.52 ± 0.77 and 1.94 ± 0.95, t-test: 12.42, p < 0.01, respectively; moderate ES [ES: 0.67, 95%CI: 0.53-0.81]). The PA, as evidenced by PAQ-A scores, significantly declined over the study period in the total sample (from 2.46 ± 0.71 to 2.25 ± 0.73, t-test: 9.34, p < 0.01; small ES (ES: 0.33]), and when observed separately for boys (from 2.75 ± 0.71 to 2.52 ± 0.77, t-test: 6.71, p < 0.01; small ES (ES: 0.24]), and for girls (from 2.14 ± 0.56 to 1.94 ± 0.55, t-test: 7.16, p < 0.01; small ES [ES: 0.25]) (Figure 2A)” (please see Statistics and Results – first paragraph). Thank you.

2.           For my surprise, the authors used logistic regression. However, logistic regression does not allow identifying either the global adjustment of performed models.  In line with that, I suggest SEM analysis (for manifest variables) or Path analysis whether models are saturated (Kline, 2016). These kinds of analysis are more robustness, especially, because authors recruited a large sample.

RESPONSE: Indeed, the specified statistics could be useful for the purpose of the study, but herein we followed the suggestions and statistical approaches from previous studies which examined similar issues in adolescence, where in most cases authors calculated logistic regression analyses for similar criteria in adolescence. We believe that it allowed us to identify both “statistical significance”, but also “relative importance” of studied predictors in explaining criterion (PA-decline/PA-incline). However, if you will insist on specified statistical procedures we will do our best and try to meet your suggestion. Thank sou.

3.           It is unclear if authors based their statistics differences on p-value or confidence interval. If authors based their interpretation on CI-95% please reported the reference, which justifies it.

RESPONSE: Indeed, in the original version of the manuscript it wasn’t clear did we rely on confidence intervals or p-levels. It is now more clearly specified for each calculation (please see Results subsection for amendments). Thank you.

4.           In line 196 (page 5), authors said the following: “A p-value of 95% was applied”. Please note that p-value is not a 95%. Otherwise, p-value represents the level of probability of rejecting the null hypothesis. In this case must be 0.05, when the confidence interval is 95%.

RESPONSE: Amended accordingly. Thank you.

Results

Results heading is well reported and described. However, based on my previous comments, they can be modifying.

RESPONSE: Thank you. Text is amended in some parts according to your suggestions

Discussion

In general the discussion is well done conducted.

RESPONSE: Thank you.

Staying at your disposal for any further changes and improvements

Authors